# Enhanced monocyte migratory activity in the pathogenesis of structural remodeling in atrial fibrillation

**Katsutoshi Miyosawa**[1,2], **Hiroshi Iwata**[1]*, **Asuka Minami-Takano**[1], **Hidemori Hayashi**[1], **Haruna Tabuchi**[1], **Gaku Sekita**[1], **Tomoyasu Kadoguchi**[1], **Kai Ishii**[1], **Yui Nozaki**[1], **Takehiro Funamizu**[1], **Yuichi Chikata**[1], **Satoshi Matsushita**[3], **Atsushi Amano**[3], **Masataka Sumiyoshi**[4], **Yuji Nakazato**[5], **Hiroyuki Daida**[1], **Tohru Minamino**[1,6]

**1** Department of Cardiovascular Biology and Medicine, Juntendo University Graduate School of Medicine, Tokyo, Japan, **2** Tokyo New Drug Research Laboratories, Kowa Company, Ltd., Tokyo, Japan, **3** Department of Cardiovascular Surgery, Juntendo University Graduate School of Medicine, Tokyo, Japan, **4** Department of Cardiovascular Medicine, Juntendo University Nerima Hospital, Tokyo, Japan, **5** Department of Cardiovascular Medicine, Juntendo University Urayasu Hospital, Chiba, Japan, **6** Japan Agency for Medical Research and Development-Core Research for Evolutionary Medical Science and Technology (AMED-CREST), Japan Agency for Medical Research and Development, Tokyo, Japan

* hiroiwata-circ@umin.ac.jp

**Data Availability Statement:** All relevant data are within the manuscript and its Supporting Information files.

## Abstract

### Background and aims

Pathophysiological roles of monocytes in atrial fibrillation (AF), particularly for the progression of structural remodeling of the left atrium (LA), remain elusive. This study examined the association between the characteristics of circulating and local monocytes and extent of structural remodeling in LA, gauged by LA size, in AF patients.

### Methods

First, 161 AF patients who were referred for catheter ablation were enrolled and divided into two groups according to the median of LA diameter (≤39 mm: normal LA group, >39 mm: enlarged LA group). As a control group, 22 patients underwent catheter ablation for paroxysmal supraventricular tachycardia (PSVT) without history of AF were analyzed. Blood samples were collected for flow cytometric analyses to evaluate monocyte subsets based on the levels of CD14 and CD16. Moreover, monocytes were isolated from blood to measure CC chemokine receptor 2 (CCR2) transcripts and protein levels, and migratory activity toward monocyte chemoattractant protein 1 (MCP-1). Second, to characterize the local monocytes in the atrial wall in AF, the resected left atrial appendages (LAA) in AF patients underwent cardiac surgery were histologically evaluated (n = 20).

### Results

The proportions of monocyte subsets based on CD14 and CD16 expressions were not significantly different between the normal and enlarged LA group. Both transcripts and total protein levels of CCR2 in monocytes were higher in the enlarged LA group compared to

**Funding:** This work was supported by a Grant for Cross-disciplinary Collaboration, Juntendo University to KM, HI, HD (30-13, https://www.juntendo.ac.jp). The funders had no role in study design, data collection and analysis, decision to publish, or preparation of the manuscript. One of the authors, Katsutoshi Miyosawa, is employed by a commercial company, Kowa Company, Ltd. The funder (Kowa Company, Ltd.) provided support in the form of salaries for author [KM], but did not have any additional role in the study design, data collection and analysis, decision to publish, or preparation of the manuscript. The specific roles of the author is articulated in the 'author contributions' section.

**Competing interests:** One of the authors [KM] received salary from the commercial company, Kowa Company, Ltd. The company did not have any additional role in the study design, data collection and analysis, decision to publish, or preparation of the manuscript. This does not alter our adherence to PLOS ONE policies on sharing data and materials.

those in the normal LA group. In the enlarged LA group, monocytes exhibited more enhanced migratory activity than the normal LA group. Moreover, we found a significantly higher number of CCR2-positive monocytes/macrophages in the LAA in the enlarged LA group.

## Conclusion

Enhanced migratory activity in circulating and local monocytes may play a pivotal role in the pathogenesis of progression in atrial remodeling in AF patients.

## Introduction

Atrial fibrillation (AF), recognized as the most common pathological cardiac arrhythmia, is associated with an increased risk of mortality and morbidities, 2-fold increased risk of mortality, and a 5-fold increased risk of thromboembolic complications [1]. Even though catheter ablation has recently become an established therapy for patients with AF, the high incidence of recurrence following the procedure still limits its merit [2, 3]. Therefore, an understanding of the detailed mechanisms of the pathophysiology of AF is clinically important for establishing both pharmacotherapeutic and interventional approaches to prevent morbidity and mortality in patients with AF. Three-dimensional (3D) structural remodeling of the atrium, such as enlargement of the atria by augmentation of fibrotic tissue, is a hallmark of AF pathogenesis [4]. In addition, the electrophysiological changes are evident due to the progressive fibrosis and perturbation of gap junction integrity [5]. Although the underlying mechanisms of these remodeling processes have not been fully elucidated, accumulating lines of evidence indicate that local inflammation in the atrium is involved in the disease pathogenesis of AF [6].

Tissue inflammation involves recruitment of circulating monocytes by chemotaxis, a major function of monocytes in the pathophysiology of various cardiovascular disorders, such as atherosclerosis, myocardial infarction, and heart failure [7]. Chemokines, such as monocyte chemoattractant protein 1 (MCP-1) and CX3C chemokine (fractalkine), are locally synthesized and released by various types of cells, including endothelia, macrophages and fibroblasts, generating gradients of those substances, which attract and recruit circulating monocytes [8]. Once monocytes adhere to endothelia, they extravasate into the subendothelium and eventually differentiate into macrophages, where they further release inflammatory cytokines and chemokines, fueling inflammation and aiding the tissue repair process. The pathological significance of chemotaxis by monocytes has been implied not only in atherosclerotic plaque formation, but also in cardiac remodeling after myocardial injury [7]; however, the role of monocyte chemotaxis in structural and electrical remodeling in the atrium, as a component of AF pathogenesis, has yet to be clarified. In this study, we therefore investigated the characteristics of monocytes, particularly focusing on monocyte chemotaxis, to examine the association with progressive structural remodeling of the left atrium in patients with AF.

## Materials and methods

### Participants

This cross-sectional observational study included two patient populations with AF and one without AF; Population 1: 22 patients without AF history, who underwent catheter ablation for paroxysmal supraventricular tachycardia (PSVT), Population 2: 161 AF patients who

underwent catheter ablation, Population 3): 20 AF patients who underwent cardiac surgeries for mitral regurgitation (n = 5), aortic stenosis (n = 5), aortic regurgitation (n = 4), combined valvular disease (n = 2), coronary artery bypass grafting (CABG) (n = 2), and atrial septal defect (n = 2). Patients in the Population 1) and 2) were underwent catheter ablation between July 2017 and May 2019, while cardiac surgeries were performed between May 2016 and February 2019 in the Population 3). All catheter ablation and cardiac surgery procedures were undergone at Juntendo University Hospital, Tokyo, Japan. In the Population 2), no patient had severe mitral regurgitation which might have substantial pathological effects to promote atrial remodeling independent from AF. In the Population 3), the left atrial appendage (LAA) was resected for preventing future thrombotic events, and subjected to immnohistochemical analysis. Echocardiography before the procedure measured parameters including left atrial diameter and left ventricular ejection fraction. This study was approved by the Institutional Review Board of Juntendo University School of Medicine and is registered in the University Hospital Medical Information Network Clinical Trial Registry (ID: UMIN000041762). Written informed consent was obtained from all participants.

## Blood collection and peripheral blood mononuclear cell isolation

Blood samples were collected before catheter ablation. Plasma was separated by centrifugation within 30 minutes after blood withdrawal. Peripheral blood mononuclear cells (PBMCs) were isolated by density gradient centrifugation (Histopaque 1077, MP Biomedicals, Santa Ana, CA, USA), and then further processed for flow cytometric analysis. In a subgroup of 73 patients, PBMCs were used to isolate monocytes for gene expression analysis. PBMCs from another subgroup of 38 patients were cryopreserved with freezing medium (Bambaker, FUJI-FILM Wako Pure Chemical, Osaka, Japan) for chemotaxis assay.

## Flow cytometry

PBMCs were first incubated with True-Stain monocyte blocker (BioLegend, San Diego, CA, USA) for 20 minutes and then stained with fluorescence-labeled antibodies against CD45, HLA-DR, CD16 and CD14 (BioLegend) for 40 minutes on ice. After washing once with FACS buffer (PBS with 2% fetal calf serum), the cells were suspended in FACS buffer with propidium iodide (Dojindo Laboratories, Kumamoto, Japan) and analyzed using an FACS Canto II flow cytometer (BD Biosciences). The gating strategy used to define the monocyte subsets is shown in S1 Fig. In 18 AF patients, CCR2 antibody (BioLegend) was also added, followed by measurement of the median fluorescence intensity (MFI) of CCR2. At least 10,000 events in a monocyte-gate were analyzed with FACSDiva software (version 6) and FlowJo software (version 7.6, BD Biosciences).

## Monocyte isolation

Pure monocyte fractions were obtained using a magnetic bead-based negative selection method (MojoSort Human Pan Monocyte Isolation Kit, BioLegend). In a pilot study involving 20 blood samples, the purity of monocytes with the method was 90.0 ± 3.4% (mean ± S.D.).

## Gene expression analysis

Purified monocytes as isolated above were dissolved in TRIzol reagent (Thermo Fisher Scientific, Waltham, MA, USA) and total RNA was extracted with a PureLink RNA Mini Kit (Thermo Fisher Scientific) and then subjected to reverse transcription with a High-Capacity cDNA Reverse Transcription Kit (Thermo Fisher Scientific). Real-time quantitative PCR

(qPCR) analysis was conducted using Prime Time qPCR Probe Assays (CCR2: Hs. PT.58.22322484, Rplp0: Hs.PT.39a.22214824, Integrated DNA Technologies, Coraville, IO, USA) and Master Mix (Integrated DNA Technologies) on a QuantStudio 3 real-time PCR system (Thermo Fisher Scientific).

## Immunoblotting

Total protein of the monocytes was extracted from residual fractions of TRIzol after RNA isolation according to a method previously reported [9]. The protein (8 µg) was separated by 4–20% Mini-PROTEAN TGX gel (Bio-Rad, Hercules, CA, USA) and transferred to PVDF membrane with Trans-Blot Turbo (Bio-Rad). The membrane was incubated with PVDF Blocking Reagent (TOYOBO, Osaka, Japan) followed by overnight incubation with anti-CCR2 antibody (Clone: D14H7, Cell Signaling Technology, Danvers, MA, USA) and anti-cyclophilin B antibody (Clone: D1V5J, Cell Signaling Technology). After the incubation with appropriate secondary antibodies conjugated with horseradish peroxidase for 1 hour, ECL Prime Western Blotting Detection Reagent (GE healthcare, Chicago, IL, USA) was applied then chemiluminescence signal was detected on FUSION SL (Analis, Namur, Belgium). Quantitative analysis of band intensity was also performed on FUSION SL.

## Chemotaxis assay

Chemotaxis assay was carried out according to BD Biosciences technical bulletin #457 with some modifications. Briefly, monocytes were purified as described above from the cryopreserved PBMCs and then suspended in cell culture medium (RPMI1640 with 0.2% bovine serum albumin). Cells were seeded at a density of 100,000 cells per well in a FluoroBlok cell culture insert (pore size 3.0 µm, Corning Inc., Corning, NY), cell culture medium with or without 5 ng/ml of CCL2 (BioLegend) was placed in the lower chamber, and then the cells were incubated for 90 minutes at 37˚C with 5% $CO_2$. The cells that had migrated and attached to the lower side of the membrane were fixed with 4% paraformaldehyde (FUJIFILM Wako Pure Chemical) and stained with 4',6-diamidino-2-phenylindole (DAPI, Thermo Fisher Scientific). Fluorescence images were obtained from five random fields of view per well at 10x objective magnification (BZ-X700, KEYENCE, Osaka, Japan) and then the nuclei were counted using Fiji software [10], representing the cell number. Cell numbers of duplicate wells in each condition were averaged and then the chemotaxis index was calculated using the following formula: Chemotaxis index = [*number of cells that migrated with CCL2*]/[*number of cells that migrated without CCL2*].

## Fluorescence immunohistochemistry of left atrial appendage

Left atrial appendage specimens were fixed in 4% paraformaldehyde, frozen in Tissue-Tek optimal cutting temperature compound (Sakura Finetek, Tokyo, Japan) and cut into 8 µm thick slices. Sections were blocked with ImmunoBlock (KAC Co., Ltd. Kyoto, Japan) for 1 hour and then treated with TrueBlack lipofuscin autofluorescence quencher (BioTium, Fremont, CA, USA) for 30 seconds before incubation in primary antibodies for CD68 (1:50 dilution, clone PG-M1, Dako, Glostrup, Denmark) and CCR2 (1:100 dilution, clone 7A7, Abcam, Cambridge, UK) overnight at 4˚C. Fluorescent-conjugated secondary antibodies (Alexa Fluor 488 anti-mouse IgG3, 1:1000 dilution, Alexa Fluor 594 anti-mouse IgG2a, 1:1000 dilution, Thermo Fisher Scientific) were applied and then the cells were incubated for 2 hours at room temperature. Sections were finally incubated with DAPI (1 µg/ml, Thermo Fisher Scientific), washed with PBS and mounted with Fluoromount-G (Thermo Fisher Scientific). Slides were examined using and LSM880 confocal microscope (Carl Zeiss, Oberkochen, Germany). The

number of CD68-positive cells (monocytes/macrophages) and CCR2-positive monocytes/macrophages in 10 random field of view images (20x objective) was counted manually in a blinded fashion.

## Statistical analysis

Statistical analyses were carried out by using GraphPad Prism version 7.02 (GraphPad Software, Inc. La Jolla, CA) and JMP 12 software (SAS institute Inc., Cary, NC). Continuous variables are expressed as the mean ± standard deviation (SD) or median values with interquartile ranges (IQR) with the results of the Shapiro-Wilk normality test. Statistical analysis of continuous variables was achieved using the unpaired t-test or non-parametric Mann-Whitney test for comparison of two groups, and one-way ANOVA followed by post hoc analysis with Tukey's post-hoc test for comparison of the three groups. Pearson's chi-squared test or Fisher's exact test was used to compare categorical variables. Differences with $p < 0.05$ were considered statistically significant.

## Results

### Background clinical demographics

The background demographics are listed and compared in Table 1 between patients with and without AF (PSVT, non-AF vs. AF), and with and without progressive left atrium structural remodeling in AF patients. Patients with AF were divided by the median of the left atrial diameter (LAD), 39 mm, as an indicator of structural remodeling. Accordingly, participants were divided into three groups; non-AF (PSVT) group (n = 22), AF with normal LA group (n = 83),

**Table 1. Clinical characteristics of the study participants.**

|  | Non-AF (PSVT) | AF | | Non-AF (PSVT) vs. AF | Among three groups | Normal LA vs. Enlarged LA |
|---|---|---|---|---|---|---|
|  | N = 22 | Normal LA | Enlarged LA |  |  |  |
|  |  | N = 83 | N = 78 |  |  |  |
| Age | 60.4 ± 8.8 | 63.1 ± 8.0 | 61.9 ± 9.0 | 0.28 | 0.37 | 0.37 |
| Gender (female, %) | 18 (81.8) | 23 (27.7) | 17 (21.8) | <0.001 | <0.001 | 0.39 |
| Body mass index (kg/m$^2$) | 23.1 ± 3.6 | 23.7 ± 2.9 | 25.8 ± 3.8 | 0.048 | <0.001 | <0.001 |
| CHADS$_2$ score | 0 (0–0.5) | 1 (0–1) | 1 (0–2) | 0.022 | 0.0084 | 0.082 |
| CHF history | 0 (0%) | 6 (7.4%) | 15 (19.5%) | 0.14 | 0.014 | 0.026 |
| Hypertension | 7 (33%) | 38 (47.0%) | 37 (48.1%) | 0.22 | 0.47 | 0.89 |
| Diabetes mellitus | 2 (9.5%) | 8 (9.9%) | 19 (24.7%) | 0.54 | 0.034 | 0.014 |
| History of stroke | 0 (0%) | 4 (4.8%) | 4 (5.1%) | 0.60 | 0.78 | 1.0 |
| Dyslipidemia | 14 (66.7%) | 48 (59.3%) | 34 (44.7%) | 0.21 | 0.087 | 0.069 |
| Smoking (past, current) | 11 (52.4%) | 46 (56.8%) | 50 (64.9%) | 0.46 | 0.44 | 0.29 |
| Left atrial diameter (mm) | 32.1 ± 3.8 | 35.1 ± 3.3 | 44.2 ± 4.1 | <0.001 | <0.001 | <0.001 |
| LVEF (%) | 68.6 ± 5.3 | 66.7 ± 7.1 | 62.6 ± 9.2 | 0.10 | 0.0017 | 0.0019 |
| hs-CRP (mg/dl) | 0.035 (0.028–0.071) | 0.040 (0.023–0.093) | 0.075 (0.036–0.16) | 0.50 | 0.040 | 0.0052 |
| White blood cells (×10$^3$/μl) | 5.5 ± 2.3 | 5.0 ± 1.1 | 5.6 ± 1.6 | 0.61 | 0.056 | 0.011 |
| Monocytes (/μl) | 300 ± 167 | 302 ± 80 | 353 ± 107 | 0.35 | 0.021 | 0.0031 |
| AF type: PAF/ PeAF | - | 61/22 | 41/37 | - | - | 0.0059 |

Data are presented as mean ± SD, median (interquartile range) or n (%). CHF: congestive heart failure, hsCRP: high sensitivity C-reactive protein, LAD: left atrial diameter, LVEF: left ventricular ejection fraction, PAF: paroxysmal atrial fibrillation, PeAF: persistent atrial fibrillation, PSVT: paroxysmal supraventricular tachycardia.

and AF with enlarged LA group (n = 78). In a comparison of AF with non-AF (PSVT) patients, AF patients had more complications, were older, more were male, and they had higher BMI and higher CHADS$_2$ scores. Echocardiography showed higher LAD in AF patients compared to non-AF patients. There was no difference in the number of circulating total white blood cells and monocytes between the groups. Serum high sensitivity C-reactive protein (hs-CRP) levels were similar in non-AF and AF patients. Moreover, within AF patients, the BMI and incidences of CHF and diabetes were higher in patients in the enlarged LA group, compared with those in the normal LA group. The number of circulating total white blood cells and monocytes and the serum hs-CRP level were significantly higher in patients in the enlarged LA group. Moreover, the proportion of persistent AF (PeAF) in the enlarged LA group was substantially higher than in the normal LA group (47.4% vs. 26.5%), possibly indicating a parallel progression of structural and electrical remodeling in AF patients.

## Proportion of CD14+CD16++ non-classical monocytes was lower in AF patients compared with non-AF control, while there was no significant difference in monocyte subsets among AF patients with or without enlarged LA

To characterize circulating monocytes in AF and non-AF patients for addressing their pathological roles in AF, flow cytometric analysis was used to classify circulating monocytes into three subsets based on cell surface CD14 and CD16 expression; CD14++CD16- ("classical"), CD14++CD16+ ("intermediate"), and CD14+CD16++ ("non-classical") [11]. Among these three subsets, CD14++CD16- classical monocytes have been reported to have the highest expression of CC chemokine receptor 2 (CCR2) and the highest migratory capacity towards the CCR2 ligand MCP-1 [12]. Compared to the non-AF group, AF groups including those with and without enlarged LA had slightly higher proportions of CD14++CD16- classical monocytes, although the difference did not reach statistical significance (Fig 1A). The

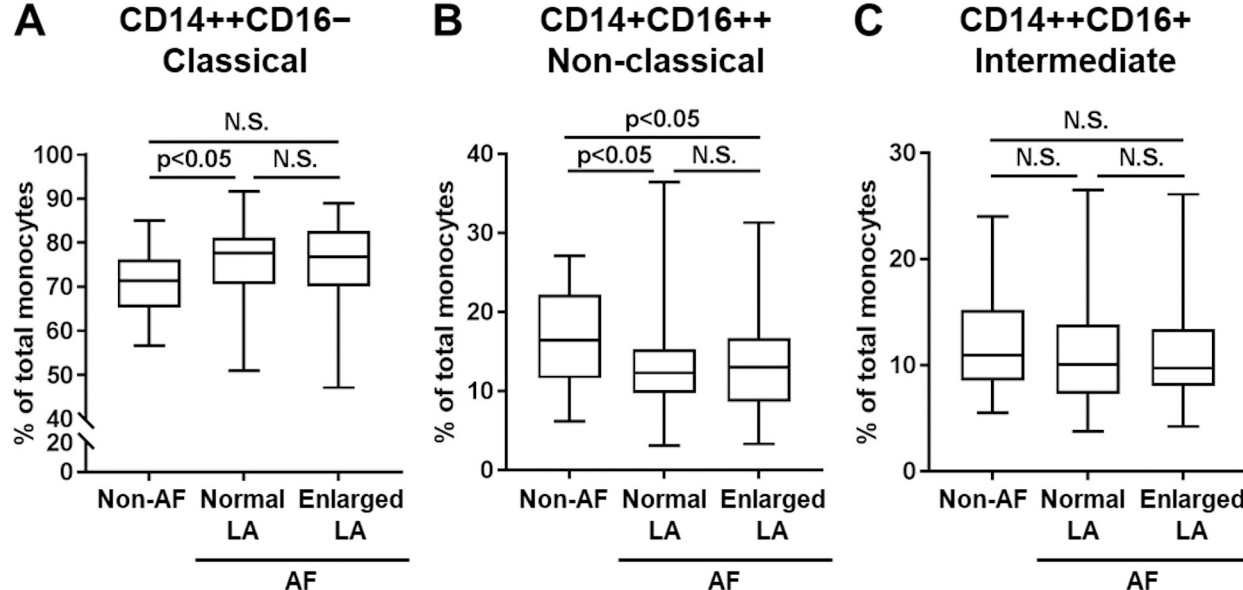

**Fig 1. Proportions of monocyte subsets in non-AF and AF patients, and AF patients subdivided into normal and enlarged LA groups.**
Percentages of CD14++CD16- classical (A), CD14+CD16++ non-classical (B) and CD14++CD16+ intermediate (C) monocyte subsets were determined in flow cytometry (non-AF: n = 22, normal LA: n = 83, enlarged LA: n = 78, Tukey's test).

proportion of CD14+CD16++ non-classical monocytes was significantly lower in AF patients compared with those in the non-AF group, while similar in with or without enlarged LA within AF patients (Fig 1B). The proportions of CD14++CD16+ intermediate monocytes were not different between AF vs. non-AF, and the normal LA vs. enlarged LA group in patients with AF (Fig 1C). These results indicated a possible association between the development of AF and the flow cytometric subpopulations of monocytes, but failed to show a significant role for the progression of LA remodeling.

## Monocyte CCR2 gene expression was higher in enlarged LA group

Chemokine receptors mediate chemotaxis of monocytes when corresponding chemokines bind to the receptors [13]. CCR2 is one of the most characterized chemokine receptors in monocytes and macrophages, and its ligand MCP-1 is capable of inducing potent monocyte migration. To investigate the possible causal relationship between the structural atrial remodeling in AF and alteration of CCR2 gene transcripts in monocytes, we isolated whole monocytes in AF patients undergoing catheter ablation and then evaluated the gene products of CCR2. We found that the CCR2 gene expression levels in monocytes derived from patients in the enlarged LA group were greater compared to those in the normal LA group (normal LA: 0.97 (0.83–1.2), enlarged LA: 1.2 (0.98–1.4), normalized mRNA levels, $p < 0.01$, Fig 2).

## CCR2 protein levels in whole cells, but not on the cell surface, were higher in circulating monocytes isolated from the enlarged LA group

Next, we attempted to measure the protein levels of CCR2 and to address the cellular localization of its protein expression in circulating monocytes. First, cell surface CCR2 protein levels

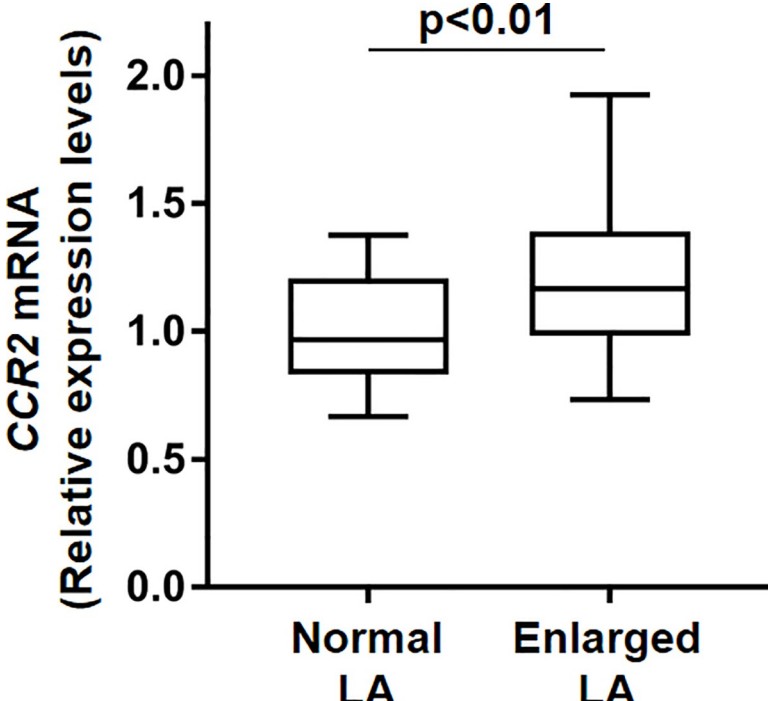

**Fig 2. Higher CCR2 mRNA levels in monocytes from AF patients in enlarged LA group.** Monocytes were isolated from PBMCs by a magnetic bead-based negative isolation method and then subjected to RNA isolation. Gene products were measured by real-time qPCR. mRNA levels of CCR2 are shown as relative to the normal LA group normalized by Rplp0 (normalLA: n = 25, enlarged LA: n = 38, unpaired t-test).

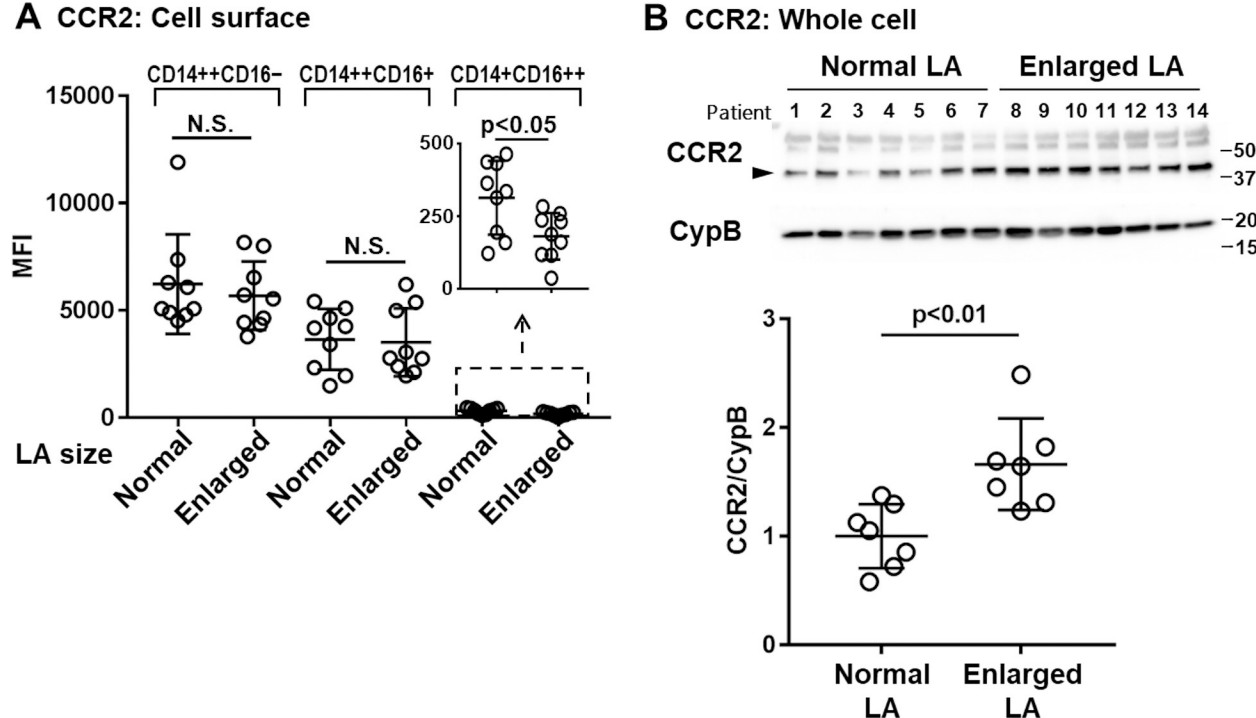

**Fig 3. Increased monocyte CCR2 protein levels in whole cells, but not on the cell surface, in patients with enlarged LA.** A) Cell surface levels of CCR2 in monocytes were determined by flow cytometry. After being classified into three subsets by flow cytometry, the median fluorescence intensity (MFI) of CCR2 was determined in each population. (normal LA: n = 9, enlarged LA: n = 9, mean ± SD, N.S.: not significant, unpaired t-test) B) Total CCR2 protein levels in monocytes were determined by Western blotting (upper panel). The arrowhead indicates the bands corresponding to CCR2. Band intensity of CCR2 was quantified and then normalized with cyclophilin B (CypB) levels (lower panel) (normal LA: n = 7, enlarged LA: n = 7, mean ± SD, unpaired t-test).

in monocytes were evaluated by flow cytometry. Consistent with previous reports,[12] expression of CCR2 was the highest in the CD14++CD16- classical monocytes subset, followed by CD14++CD16+ intermediate monocytes and CD14+CD16++ non-classical monocytes (Fig 3A). In the CD14++CD16- classical and CD14++CD16+ intermediate subsets, there was no difference in CCR2 expression on the cell surface between the normal and enlarged LA groups, while that on the cell surface of CD14+CD16++ non-classical monocytes was significantly lower in the enlarged LA group, although the expression of CCR2 in CD14+CD16++ non-classical monocytes was substantially lower (approximately 1/20 compared to CD14++CD16- classical monocytes). Next, we measured CCR2 protein levels in whole cell lysates of unfractionated monocytes by Western blotting (Fig 3B upper panel). Quantification analysis revealed significantly elevated CCR2 protein levels in the enlarged LA group compared to those in the normal LA group (normal LA: 0.46 ± 0.14, enlarged LA: 0.77 ± 0.19, relative to cyclophilin B, p<0.01, Fig 3B lower panel). Taken together, these results indicate that the increase in CCR2 transcripts in monocytes (Fig 2) is reflected as increased CCR2 protein levels in the cytosol but not as an increased cell surface protein levels.

## Monocytes derived from AF patients in the enlarged LA group showed higher migratory capacity

To investigate whether monocyte migratory activity is associated with the progression of atrial structural remodeling, we measured the migratory activity of circulating monocytes toward

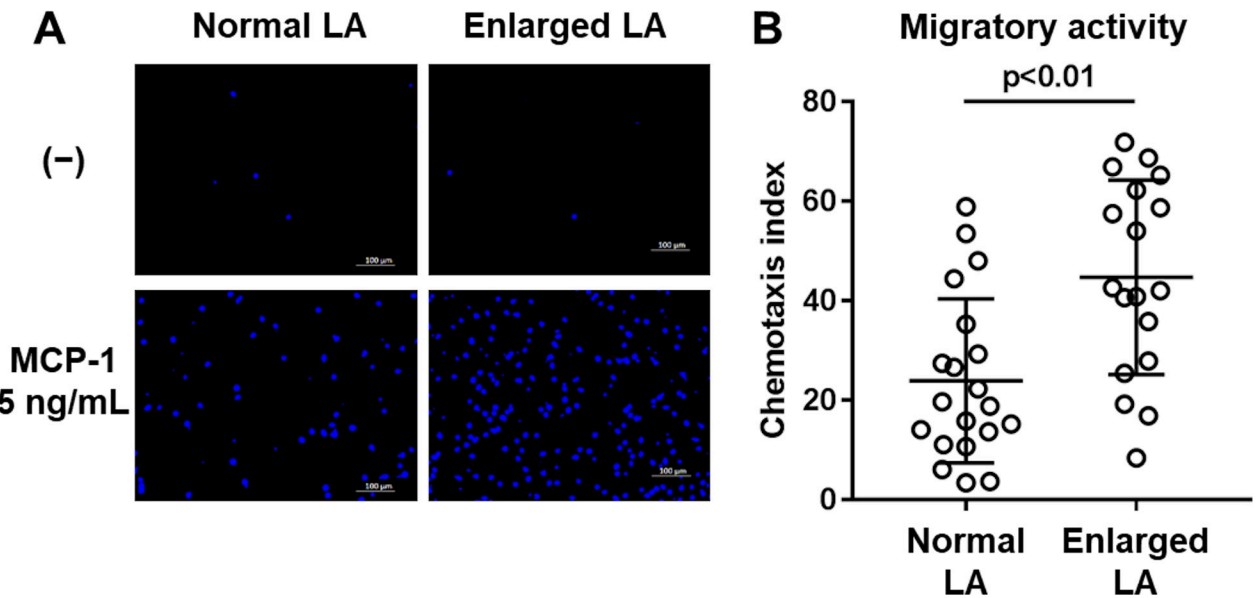

**Fig 4. Enhanced migratory activity of monocytes toward MCP-1 in patients with enlarged LA.** Monocytes were isolated and subjected to migratory activity assay. A) Representative fluorescence images of migrated cells after incubation with or without 5 ng/ml of MCP-1 for 90 minutes (Scale bar: 100 μm, 20x objective). B) Number of nuclei was counted in fluorescence images obtained from five random fields of view per well at 10x objective magnification. Chemotaxis index was calculated as described in the Methods section (normal LA: n = 20, enlarged LA: n = 18, mean ± SD, unpaired t-test).

MCP-1, a chemoattractant and a ligand for CCR2, by using a modified Boyden-chamber assay. The number of monocytes that migrated to 5 ng/ml of MCP-1 was higher in the enlarged LA group (Fig 4A). The migratory activity of monocytes, expressed as the chemotaxis index [14], was significantly higher in the enlarged LA group (n = 18) compared to the normal LA group (n = 20) (normal LA: 23.9 ± 16.5, enlarged LA: 44.7 ± 19.5, p<0.01, Fig 4B). These results suggest that circulating monocytes in AF patients with advanced LA structural remodeling have a higher migratory capacity toward MCP-1.

### Increased number of CCR2-positive monocytes/macrophages in the wall of left atrial appendages in the enlarged LA group

To clarify the possible association between the enhanced migratory capacity of monocytes with the pathology of left atrial remodeling in AF, we performed immunohistochemical analysis of left atrial appendages (LAA) obtained from AF patients who underwent cardiac surgery for mitral regurgitation, aortic stenosis, aortic regurgitation, combined valvular disease, CABG, or atrial septal defect with LAA excision (n = 20). There was no significant difference between the two groups (LAD ≤ or >39 mm) except larger BMI in the enlarged LA group (Table 2). Immunofluorescence staining of LAA wall detected monocytes/macrophages with CD68 expression, some of which expressed CCR2 (Fig 5A). We then quantified the average number of CD68+ and CCR2+ monocytes/macrophages in 10 random field of view images (20x objective) of LAA sections. In LAA of AF patients in the enlarged LA group, there were higher numbers of CD68+ monocytes/macrophages compared to those in the normal LA group (Fig 5B and 5C left panel). Furthermore, the number of CCR2-positive monocytes/macrophages was also high in LAA resected from patients with enlarged LA, suggesting the possibility of enhanced infiltration of circulating monocytes into the left atrium, which subsequently differentiated into macrophages in AF patients with advanced atrial remodeling

**Table 2. Clinical characteristics of the participants for histological analysis of left atrial appendages.**

| | Normal LA | Enlarged LA | p value |
| --- | --- | --- | --- |
| | N = 10 | N = 10 | |
| Age | 67.7 ± 6.7 | 67.4 ± 3.9 | 0.90 |
| Gender (female, %) | 3 (30) | 3 (30) | 1.0 |
| Body mass index (kg/m$^2$) | 22.2 ± 3.9 | 26.9 ± 3.1 | 0.0084 |
| CHADS$_2$ score | 1 (1–1.25) | 1.5 (0.75–2.25) | 0.45 |
| History of CHF | 8 (80%) | 3 (30%) | 0.070 |
| Hypertension | 2 (20%) | 5 (50%) | 0.35 |
| Diabetes mellitus | 1 (10%) | 3 (30%) | 0.58 |
| History of stroke | 0 (0%) | 2 (20%) | 0.47 |
| Dyslipidemia | 4 (40%) | 5 (50%) | 1.0 |
| Smoking (past, current) | 3 (30%) | 6 (60%) | 0.37 |
| Left atrial diameter (mm) | 35.9 ± 3.2 | 51.5 ± 6.8 | <0.001 |
| LVEF (%) | 66.0 ± 9.0 | 61.0 ± 4.4 | 0.64 |
| CRP (mg/dl) | 0.085 (0.0–0.22) | 0.10 (0.10–0.21) | 0.36 |
| White blood cells (×10$^3$/μl) | 5.1 ± 1.4 | 4.8 ± 1.0 | 0.56 |
| AF type (paroxysmal/ persistent/ permanent) | 5/5/0 | 2/7/1 | 0.35 |
| Surgical procedures | | | 0.70 |
| Mitral regurgitation | 3 | 2 | |
| Aortic stenosis | 2 | 3 | |
| Aortic regurgitation | 2 | 2 | |
| Combined valvular disease | 1 | 1 | |
| Coronary artery bypass grafting | 0 | 2 | |

Data are presented as mean ± SD, median (interquartile range) or n (%). CHF: congestive heart failure, CRP: C-reactive protein, LAD: left atrial diameter, LVEF: left ventricular ejection fraction.

(Fig 5B and 5C mid panel). Additionally, the ratio of CCR2+ to whole CD68+ monocytes/macrophages was slightly higher in the enlarged LA group (Fig 5C right panel), although the difference was not statistically significant.

## Discussion

In this study, we evaluated the functional differences of circulating monocytes, focusing particularly on their migratory activity, in AF patients who underwent catheter ablation with or without advanced LA structural remodeling gauged by echocardiography-measured LAD. In the monocyte isolation, we employed the magnet-based negative selection method to collect cells without any stimulation, since in the positive selection method the phenotype of monocytes may be altered by binding to CD14 antibody [15, 16]. Moreover, we found that gene products and protein levels of CCR2 in isolated monocytes were higher in AF patients whose LAD was larger than 39 mm. Circulating monocytes isolated from AF patients with enlarged LA exhibited enhanced migratory capacity to the CCR2 ligand MCP-1. Finally, histological analysis in LAA surgically resected from AF patients confirmed that the number of CCR2-positive monocytes/macrophages was higher in AF patients with enlarged LA.

In the present study, CCR2 transcripts and total CCR2 protein levels, but not cell surface CCR2 protein levels, seem to be associated with the migratory activity of monocytes towards MCP-1, suggesting that cell surface chemokine receptor levels do not necessarily account for the migratory activity. Similar to our findings, enhanced migratory activity of PBMCs towards

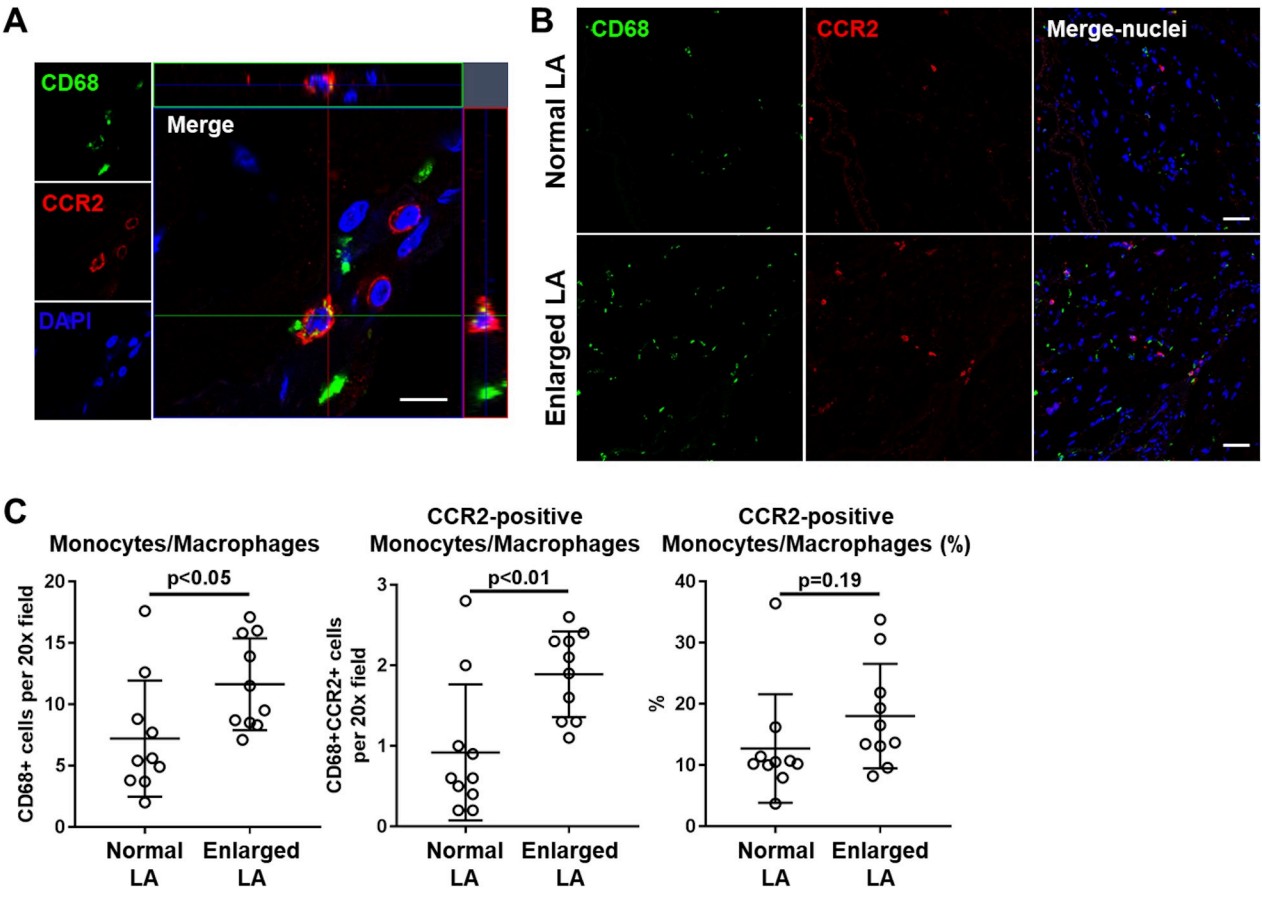

**Fig 5. Increased number of monocytes/macrophages in the wall of left atrial appendages in AF patients with enlarged LA.** Immunofluorescence staining of left atrial appendages of AF patients who underwent cardiac surgery. A) Representative image of CD68 (green) and CCR2 (red) double-positive cells (monocytes/macrophages) at a high-power field magnification (63x objective) in a left atrial appendage. Scale bar: 5 μm. B) Representative images at low-power magnification (20x objective) in normal and enlarged LA patients. Scale bar: 10 μm. C) The number of CD68-positive cells (monocytes/macrophages) and CCR2-positive monocytes/macrophages, and percentage of CCR2-positive macrophages per randomly selected field of view at 20x objective magnification (normal LA: n = 10, enlarged LA: n = 10, mean ± S.D., unpaired t-test).

CCL5 was observed in chronic obstructive pulmonary disease (COPD) patients compared to those of control subjects (nonsmokers and smokers), while there was no difference in cell surface levels of CCR5 [17]. Another study on CCR7 showed that surface CCR7 levels on B cells measured by flow cytometry did not differ between mutated chronic lymphocytic leukemia (M-CLL) versus un-mutated CLL (U-CLL) patients, while the transcripts and total CCR7 protein levels were significantly increased in U-CLL cells compared to those in M-CLL [14]. The enhanced migratory activity of B cells to CCL21 were manifested in U-CLL, which was explained by efficient recycling of CCR7 [14]. Cell migration through chemokine receptors is tightly regulated by receptor desensitization, internalization and recycling for example. Further investigation is needed to clarify whether alteration in the subcellular distribution of CCR2 underlies the enhanced migratory activity of monocytes in AF patients.

Recruitment of inflammatory monocytes has been studied in the pathogenesis of atherosclerosis; however, recent studies revealed that monocytes with CCR2 expression are recruited not only to atherosclerotic plaque but also to ventricular myocardium. Distinct populations of CCR2- and CCR2+ macrophages were found in human myocardium, where CCR2- macrophages are replenished mainly by proliferation, while CCR2+ macrophages are maintained by

both monocyte recruitment and proliferation [18]. In a mouse study, expression of chemokines or cytokines is elevated in CCR2+ macrophages after myocardial injury, indicating recruitment of monocytes to the injured site [19]. Previously, an investigation of macrophages was conducted in LAA of AF patients who underwent cardiac surgery, and showed that accumulation of macrophages in LAA wall was well correlated with left atrial diameter [20], although the subtypes of macrophages and cell surface antigens were not clarified. Our immunohistochemical analysis revealed that higher number of CCR2-positive and CCR2-negative monocytes/macrophages in LAA wall in AF patients with progressive atrial remodeling, which could possibly indicate enhanced infiltration of monocytes/macrophages. Infiltrated monocytes may differentiate into macrophages and then activate the pro-inflammatory process by releasing cytokines or chemokines for further recruitment of monocyte/macrophage lineage cells or induce reparative function by facilitating tissue fibrosis. A novel role of macrophages was found in the distal atrioventricular node in humans and mice, where macrophages couple to cardiomyocytes via connexin-43, thereby affecting cardiac conduction [21]. Although such findings may represent the physiological activity of resident macrophages in cardiac tissue, infiltration of circulating monocytes and subsequent differentiation into macrophages in the atrial wall may also underlie the electrical remodeling process by altering the electric conductance and generating reentrant circuits [22].

Inflammatory biomarkers in blood, such as interleukin-6 and MCP-1, have been examined in AF patients; however, the relation between plasma MCP-1 levels and AF remains controversial. Plasma MCP-1 levels in patients with lone AF were reported to be higher compared to those in control subjects [23]; however, a recent genetic analysis reported that genetic predisposition to higher MCP-1 levels was not associated with atrial fibrillation [24]. It is noteworthy that the degree of fibrosis in atrial myocardium was correlated with MCP-1 levels in epicardial adipose tissue dissected from left atrial appendages in patients with AF [25]. suggesting that tissue local chemokine expression mainly contributes to atrial remodeling of the left atrium and clinical AF progression. Our study showed no significant differences in plasma MCP-1 levels between non-AF and AF patients, nor between normal and enlarged LA (S2 Fig). Given the presumably short half-life of MCP-1 in blood and the importance of local production of chemokines [26], it may be reasonable to assume that elevated blood MCP-1 levels could not be detected in the enlarged LA group.

A previous report by Suzuki *et al.* indicated an increased proportion of the CD14++CD16 + intermediate monocyte subset in AF subjects compared to that of healthy control subjects [27], while our flow cytometric analysis showed no difference in the proportions of intermediate monocytes between non-AF and AF participants. This discrepancy in the results between the two studies may arise from inclusion of a control group population; in the Suzuki *et al.*'s study, healthy individuals without any cardiovascular health problems and who were supposedly low in inflammation served as the control group, whereas in our study, the control group was comprised of PSVT patients who could be predisposed with inflammatory status [28]. Meanwhile, consistent with our results, Suzuki *et al.* also demonstrated that there was no association between LA dimension and the proportions of monocyte subsets [27].

AF incidence is increased with age. Also male was reported to be more susceptible to AF, although the lifetime risk of AF was not different between male and female [29]. In our study, there was no gender- and age-related difference in characteristics of circulating monocytes in AF patients who underwent catheter ablation (data not shown). Studies in healthy subjects have suggested age- and gender-related changes of monocyte characteristics such as monocyte subsets and CCR2 levels in monocytes [30, 31]; however, more comprehensive analyses are necessary to fully understand the role of monocyte migration in AF pathogenesis.

Diabetes is associated with not only increased number of circulating monocytes [32], but also activation of inflammatory monocytes and macrophages [33]. Moreover, a study enrolling patients with severe heart failure suggested a negative correlation between left ventricular ejection fraction (LVEF) and the abundance of CCR2-positive macrophage in the cardiac tissue [18]. In our study, incidences of diabetes and heart failure were higher and LVEF was lower in patients with enlarged LA compared with normal LA group, indicating that these factors might be directly or indirectly associated with enhanced CCR2 levels and migratory activity in circulating monocytes. Additionally, CCR2 levels and chemotaxis activity had been reported to be significantly elevated in circulating monocytes in obese subjects [34]. As BMI was higher in the enlarged LA group in both analyses of circulating and local monocytes/macrophages, higher BMI might be another possible cofounder in this study.

## Study limitations

Our study has several limitations. First, this was an exploratory study conducted in a single center and included only a small number of AF patients who underwent catheter ablation. Therefore, we may not be able to apply our results to the general AF population. Second, our data showed that monocytes in AF patients with enlarged LA exhibited enhanced migratory capacity. However, we were unable to clarify whether this was a cause or result of advanced structural remodeling of LA, so a prospective study may need to be conducted. Third, direct evidence may be obtained by animal models to further strengthen our hypothesis by using monocyte/macrophage-specific CCR2 knock-out animals. Although AF has been previously studied in some animal models, the translatability of those animal models to human AF pathogenesis is still uncertain [35]. Choosing or developing a right AF model and study design would provide highly translatable data and may shed light on the contribution of monocyte migration in the atrial remodeling process. Forth, compared to AF patients with catheter ablation whose circulating monocytes were examined in this study (Table 1), the population served for the histological analysis underwent cardiac surgeries had further complicated background (Table 2). Accordingly, the roles and the characteristics of monocytes/macrophages in the pathogenesis of atrial structural remodeling might not be completely equal in these two populations of AF. Fifth, this study used echocardiography-measured LA dimension to represent the degree of structural remodeling in AF. However, since LA volume (LAV) or volume index (LAVI) and LA maximal area might be further advisable as indicators of remodeling in AF patients,[36] future studies by using these parameters might be warranted.

## Conclusions

Enhanced migratory activity of circulating monocytes and increased number of monocytes/macrophages in the atrial wall may underlie LA remodeling and the pathophysiology of AF. Extensive investigation of monocyte biology associated with disease pathogenesis might lead to new therapeutic approaches which would contribute to reducing AF morbidity and mortality.

## Supporting information

**S1 Fig. Flow cytometry gating strategy to analyze monocyte subsets.** Doublets were excluded with forward scatter (FSC)-A versus FSC-H plots, then gated on cells including monocytes based on FSC / side scatter (SSC) plots. CD45 positive, propidium iodide (PI) negative and HLA-DR positive population was further selected. After gating-out CD14-CD16- cells, monocytes were classified into three subsets.
(DOCX)

**S2 Fig. Circulating levels of MCP-1.** Serum MCP-1 levels were determined by human MCP-1 DuoSet ELISA (R&D, Minneapolis, MN). N.S.: not significant (non-AF: n = 21, normal LA: n = 80, enlarged LA: n = 76, Tukey's test).
(DOCX)

**S1 Raw images. Original blot images.**
(PDF)

## Acknowledgments

The authors are grateful to Dr. Nozomi Hayashiji, Ms. Emiko Nakamura and Mr. Norihiko Yokoyama for their excellent technical assistance. The authors also thank the Laboratory of Molecular and Biochemical Research, and the Laboratory of Morphology and Image Analysis, Research Support Center, Juntendo University Graduate School of Medicine for their valuable technical assistance.

## Author Contributions

**Conceptualization:** Katsutoshi Miyosawa, Hiroshi Iwata.

**Data curation:** Katsutoshi Miyosawa, Hiroshi Iwata, Asuka Minami-Takano, Tomoyasu Kadoguchi, Kai Ishii, Yui Nozaki, Takehiro Funamizu, Yuichi Chikata, Satoshi Matsushita.

**Formal analysis:** Katsutoshi Miyosawa, Hiroshi Iwata, Asuka Minami-Takano, Tohru Minamino.

**Funding acquisition:** Katsutoshi Miyosawa, Hiroshi Iwata, Hiroyuki Daida.

**Investigation:** Katsutoshi Miyosawa, Hiroshi Iwata, Asuka Minami-Takano, Hidemori Hayashi, Haruna Tabuchi, Gaku Sekita, Tomoyasu Kadoguchi, Kai Ishii, Yui Nozaki, Takehiro Funamizu, Yuichi Chikata, Satoshi Matsushita.

**Project administration:** Katsutoshi Miyosawa, Hiroshi Iwata.

**Supervision:** Hiroshi Iwata, Atsushi Amano, Masataka Sumiyoshi, Yuji Nakazato, Hiroyuki Daida, Tohru Minamino.

**Visualization:** Katsutoshi Miyosawa, Hiroshi Iwata.

**Writing – original draft:** Katsutoshi Miyosawa, Hiroshi Iwata.

**Writing – review & editing:** Katsutoshi Miyosawa, Hiroshi Iwata, Asuka Minami-Takano, Hidemori Hayashi, Haruna Tabuchi, Gaku Sekita, Tomoyasu Kadoguchi, Kai Ishii, Yui Nozaki, Takehiro Funamizu, Yuichi Chikata, Satoshi Matsushita, Atsushi Amano, Masataka Sumiyoshi, Yuji Nakazato, Hiroyuki Daida, Tohru Minamino.

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
