## [Decision Letter · Decision Letter 0]

19 Aug 2020

PONE-D-20-13849

Enhanced monocyte migratory activity in the pathogenesis of structural remodeling in atrial fibrillation

PLOS ONE

Dear Dr. Iwata,

Thank you for submitting your manuscript to PLOS ONE. After careful consideration, we feel that it has merit but does not fully meet PLOS ONE’s publication criteria as it currently stands. Therefore, we invite you to submit a revised version of the manuscript that addresses the points raised during the review process.

ACADEMIC EDITOR: Please add the few comments of both reviewers to the manuscript 

We look forward to receiving your revised manuscript.

Kind regards,

Eliseo A Eugenin, Ph.D.

Academic Editor

PLOS ONE

Journal Requirements:

2.PLOS ONE now requires that authors provide the original uncropped and unadjusted images underlying all blot or gel results reported in a submission’s figures or Supporting Information files. This policy and the journal’s other requirements for blot/gel reporting and figure preparation are described in detail at https://journals.plos.org/plosone/s/figures#loc-blot-and-gel-reporting-requirements and https://journals.plos.org/plosone/s/figures#loc-preparing-figures-from-image-files. When you submit your revised manuscript, please ensure that your figures adhere fully to these guidelines and provide the original underlying images for all blot or gel data reported in your submission. See the following link for instructions on providing the original image data: https://journals.plos.org/plosone/s/figures#loc-original-images-for-blots-and-gels.

3.Thank you for stating the following in the Competing Interests section:

[This work was supported by a Grant for Cross-disciplinary Collaboration, Juntendo University to KM, HI, HD (30-13, https://www.juntendo.ac.jp). The funders had no role in study design, data collection and analysis, decision to publish, or preparation of the manuscript.].   

We note that one or more of the authors are employed by a commercial company: Tokyo New Drug Research Laboratories, Kowa Company, Ltd.

Additional Editor Comments (if provided):

Dear Dr. Iwata

Thank you for your patience and our apologies for the extended time for the review process. Please modify the manuscript as requested and send it back

Best Regards

Eliseo Eugenin

Reviewers' comments:

Reviewer's Responses to Questions

**Comments to the Author**

1. Is the manuscript technically sound, and do the data support the conclusions?

Reviewer #1: Yes

Reviewer #2: Yes

2. Has the statistical analysis been performed appropriately and rigorously? 

Reviewer #1: Yes

Reviewer #2: Yes

3. Have the authors made all data underlying the findings in their manuscript fully available?

Reviewer #1: Yes

Reviewer #2: Yes

4. Is the manuscript presented in an intelligible fashion and written in standard English?

Reviewer #1: Yes

Reviewer #2: Yes

5. Review Comments to the Author

Reviewer #1: Miyosawa et al present a manuscript for review in which they performed a cross-sectional study of patients with atrial fibrillation referred for catheter ablation. They performed flow cytometry on blood samples to evaluate monocyte subsets and evaluated chemokine receptor 2 transcripts and protein levels, as well as migratory activity of the monocytes. In a subset of patients who underwent cardiac surgery who had left atrial appendage excision, they also evaluated histological monocyte characteristics in the atrial wall. In separating the patients based on median left atrial diameter, they found higher CCR2 levels and monocyte migratory activity in those individuals with dilated left atria.

The study is interesting and carries a clear mechanistic message for understanding of the progression/sequelae of atrial fibrillation, as well as potential targets for future therapies. The paper is well written, the findings not overstated, and most of the limitations acknowledged.

A few items to consider:

1. The abstract does not make it clear the patient population. Would mention that these are patients who were referred for catheter ablation

2. In the patients who had LAA excision, when were these surgeries performed relative to the catheter ablation?

3. Left atrial diameter is a one dimensional measurement and may be subject to significant error. The authors acknowledge that atrial fibrillation induces 3-dimensional remodeling and enlargement of the left atrium. Did the authors look at the echocardiographic data to attempt additional left atrial analysis such as LA maximal area, LA indexed volume, or functional parameters (e.g. reservoir volume, etc.)? If collected, it would make for an interesting sensitivity analysis; i.e. whether stratifying patients by median LA indexed volume would produce similar results. If this data was not collected, would add that as a limitation of the study

4. Would consider the different rates of heart failure in the enlarged vs non-enlarged LA groups as a potential confounder. The authors cite evidence that chemotaxis is abnormal in patients with heart failure, and it is well known that atrial fibrillation and congestive heart failure have a complex bidirectional relationship. This might be mentioned in the discussion and/or as a limitation of the study.

Reviewer #2: Summary: Iowata et al examine the association of monocytes was structural remodeling of the left atrium, by examining blood samples from hundred 61 patients with atrial fibrillation. There a specific examination of chemo kind receptor protein levels, and chemoattractant related to monocytes. The authors conclude from the data shown that patients with enlarged left atria have higher level of monocytes/macrophages, and this presence may lead to higher rates of progression of atrial remodeling in this patient population. Overall, the rationale of the study is sound, the manuscript is well-written, and techniques and experimental approach are very thorough.

Questions and comments:

1. Regarding the type of specimen procured and examined: were there any potential confounding factors, such as any prior incidents of myocarditis or other forms of inflammation that would have altered monocyte/macrophage levels?

2. Whether any sex-related differences in monocyte/chemo kind levels, considering the fact that there was a much higher proportion of atrial fibrillation patients were not only men, but also older?

3. Some of the potential confounding factors between comparing patients with atrial fibrillation versus non-AF could include diabetes mellitus (in and of itself and inflammatory state that could affect monocyte/macrophage levels independent of atrial fibrillation risk). How is this and other factors taken into account when interpreting the findings?

4. In Figure 5: please clarify how you elucidated that the infiltrating cells were monocytes, and subsequently differentiated into macrophages following infiltration. The sentence beginning on line 320 implies that is the case; if it is not the case, please reword the sentence to reflect the findings more accurately if needed.

5. On line 331, please verify magnification – is it supposed to be 63x? It is listed as 630x.

6. PLOS authors have the option to publish the peer review history of their article (what does this mean?). If published, this will include your full peer review and any attached files.

Reviewer #1: No

Reviewer #2: No

---

## [Author Response · Author response to Decision Letter 0]

25 Sep 2020

Response to Reviewers

The authors sincerely thank the Editors and the Reviewers for the opportunity to revise our study, as well as a number of insightful and helpful comments and suggestions, which greatly help to add values of our study. In the following response to reviewers, comments by the Reviewer were in bold and changes to the manuscript have been indicated by responses in red font.

Reviewer #1: Miyosawa et al present a manuscript for review in which they performed a cross-sectional study of patients with atrial fibrillation referred for catheter ablation. They performed flow cytometry on blood samples to evaluate monocyte subsets and evaluated chemokine receptor 2 transcripts and protein levels, as well as migratory activity of the monocytes. In a subset of patients who underwent cardiac surgery who had left atrial appendage excision, they also evaluated histological monocyte characteristics in the atrial wall. In separating the patients based on median left atrial diameter, they found higher CCR2 levels and monocyte migratory activity in those individuals with dilated left atria.

The study is interesting and carries a clear mechanistic message for understanding of the progression/sequelae of atrial fibrillation, as well as potential targets for future therapies. The paper is well written, the findings not overstated, and most of the limitations acknowledged.

[Response]

All authors thank the Reviewer for his/her positive and constructive comments on our manuscript. We have carefully revised the manuscript in accordance with the Reviewer’s comments and suggestions point by point, which have greatly helped to increase the scientific value of our study.

A few items to consider:

1. The abstract does not make it clear the patient population. Would mention that these are patients who were referred for catheter ablation

[Response]

We thank and agree with the Reviewer for the suggestion that patient populations in the abstract should be more accurate and clearer. Accordingly, we have revised the abstract for clarifying we evaluated three different populations in this study, including 1) patients underwent catheter ablation for paroxysmal supraventricular tachycardia (PSVT) without history of AF as a control group, 2) patients with AF who underwent catheter ablation (n=161), and 3) patients with AF who underwent cardiac surgery whose left atrial appendages (LAA) were resected (n=20).

Revised Abstract Methods (page 2, line 32-35)

First, 161 AF patients who were referred for catheter ablation were enrolled and divided into two groups according to the median of LA diameter (≤39 mm: normal LA group, >39 mm: enlarged LA group). As a control group, 22 patients underwent catheter ablation for paroxysmal supraventricular tachycardia (PSVT) without history of AF were analyzed.

Revised Abstract Methods (page 2, line 39-41)

Second, to characterize the local monocytes in the atrial wall in AF, the resected left atrial appendages (LAA) in AF patients underwent cardiac surgery were histologically evaluated (n=20).

Revised Abstract Methods (page 2, line 42-43)

Results: The proportions of monocyte subsets based on CD14 and CD16 expressions were not significantly different between the normal and enlarged LA group.

2. In the patients who had LAA excision, when were these surgeries performed relative to the catheter ablation?

[Response]

We would like to apologize for the confusion regarding patient populations we have analyzed in this study. As mentioned in response for the first suggestion by the Reviewer, we have analyzed three different populations, 1) patients underwent catheter ablation for PSVT without history of AF as a control group, 2) AF patients undergone catheter ablation (n=161, Table 1), and 3) AF patients who undergone cardiac surgeries (listed in Supplementary Table 1 in the original manuscript) and were resected LAA for prevention of future embolic stroke (n=20). In the first and second populations (Populations 1) and 2)), we have analyzed the monocyte subsets in PBMC by flow-cytometry. Moreover, in the second population (Population 2)), to examine the association between characteristics of circulating monocytes and the degree of atrial structural remodeling, we have investigated multiple aspects of circulating monocytes including CCR2 mRNA and protein expression, and migratory activity. In the third population (Population 3)), monocyte/ macrophage lineage CD68+ cells localized in the atrial tissue were histologically characterized. There was no patient who belonged both second and third populations. The pathogenesis and the roles of monocyte/macrophage lineage cells in the two populations (Populations 2) and 3)) might be different due to more complicated background in the third population (Population 3)), although the comparison of atrial tissues between the two groups is unrealistic as it is not technically possible to obtain atrial tissue through catheter ablation procedure in Population 2). We therefore revised the Participants in the Material and Methods section and added the following new sentences in the limitation in the Discussion section, and Supplementary Table 1 was moved to Table 2 in the revised manuscript.

Revised Participants section in the Material and Methods (page 5, line 85-99):

This cross-sectional observational study included two patient populations with AF and one without AF; Population 1): 22 patients without AF history, who underwent catheter ablation for paroxysmal supraventricular tachycardia (PSVT), Population 2): 161 AF patients who underwent catheter ablation, Population 3): 20 AF patients who underwent cardiac surgeries for mitral regurgitation (n=5), aortic stenosis (n=5), aortic regurgitation (n=4), combined valvular disease (n=2), coronary artery bypass grafting (CABG) (n=2), and atrial septal defect (n=2). Patients in the Population 1) and 2) were underwent catheter ablation between July 2017 and May 2019, while cardiac surgeries were performed between May 2016 and February 2019 in the Population 3). All catheter ablation and cardiac surgery procedures were undergone at Juntendo University Hospital, Tokyo, Japan. In the Population 2), no patient had severe mitral regurgitation which might have substantial pathological effects to promote atrial remodeling independent from AF. In the Population 3), the left atrial appendage (LAA) was resected for preventing future thrombotic events, and subjected to immnohistochemical analysis. Echocardiography before the procedure measured parameters including left atrial diameter and left ventricular ejection fraction. 

Revised Study limitation (page 22, line 460-465):

Forth, compared to AF patients with catheter ablation whose circulating monocytes were examined in this study (Table 1), the population served for the histological analysis underwent cardiac surgeries had further complicated background (Table 2). Accordingly, the roles and the characteristics of monocytes/macrophages in the pathogenesis of atrial structural remodeling might not be completely equal in these two populations of AF. 

3. Left atrial diameter is a one dimensional measurement and may be subject to significant error. The authors acknowledge that atrial fibrillation induces 3-dimensional remodeling and enlargement of the left atrium. Did the authors look at the echocardiographic data to attempt additional left atrial analysis such as LA maximal area, LA indexed volume, or functional parameters (e.g. reservoir volume, etc.)? If collected, it would make for an interesting sensitivity analysis; i.e. whether stratifying patients by median LA indexed volume would produce similar results. If this data was not collected, would add that as a limitation of the study.

[Response]

We greatly appreciate the opportunity to clarify this important point. In light of the insightful suggestion by the Reviewer, which we completely agree with, we have tried to obtain left atrial volume (LAV) by re-reviewing echocardiographic data. However, unfortunately, it was unavailable in the substantial number of participants, which may cause underpowered analyses. Accordingly, in the Study limitation section of the revised manuscript, we have mentioned the importance to use LAV, LAVI or LA maximal area which are the better indicator of 3D structural remodeling, and the need for future study of AF using these parameters.

Revised Study limitation (page 22, line 465-469):

Fifth, this study used echocardiography-measured LA dimension to represent the degree of structural remodeling in AF. However, since LA volume (LAV) or volume index (LAVI) and LA maximal area might be further advisable as indicators of remodeling in AF patients (Lang RM. et al., Eur Heart J Cardiovasc Imaging 2015;16:233-70), future studies by using these parameters might be warranted.

4. Would consider the different rates of heart failure in the enlarged vs non-enlarged LA groups as a potential confounder. The authors cite evidence that chemotaxis is abnormal in patients with heart failure, and it is well known that atrial fibrillation and congestive heart failure have a complex bidirectional relationship. This might be mentioned in the discussion and/or as a limitation of the study.

[Response]

The Reviewer raised a very important point. In this study, the proportion of patients with history of congestive heart failure (CHF) is significantly higher in the enlarged LA group compared to that in the non-enlarged LA group (Table 1, p<0.026). As noted by the Reviewer, atrial fibrillation and heart failure often complicated each other (Kotecha D. et al., Eur Hear J 2015;36:3250), which may suggest the shared underling mechanisms in the pathogenesis of the two diseases. A previous study showed the negative correlation between abundance of CCR2-positive macrophages in the left ventricular (LV) tissue in patients with heart failure and LV systolic function, suggesting the pathological roles of CCR2-positive macrophages to suppress cardiac function in heart failure (Bajpai G. et al., Nat Med 2018;24:1234-1245). In our study, ejection fraction in the enlarged LA group was significantly lower than non-enlarged group (p=0.0019), which might indicate the association between enhanced monocyte/macrophage migratory activity in patients with reduced LV function. We believe this is a very important discussion; therefore, we have included new Discussion in the revised manuscript as follows.

Revised Discussion (page 21, line 436-446)

Diabetes is associated with not only increased number of circulating monocytes (Min D. et al., Mediators Inflamm 2012;2012:649083), but also activation of inflammatory monocytes and macrophages (Flynn MC. et al., Front Pharmacol 2019;10:666). Moreover, a study enrolling patients with severe heart failure suggested a negative correlation between left ventricular ejection fraction (LVEF) and the abundance of CCR2-positive macrophage in the cardiac tissue (Bajpai G. et al., Nat Med 2018;24:1234). In our study, incidences of diabetes and heart failure were higher and LVEF was lower in patients with enlarged LA compared with normal LA group, indicating that these factors might be directly or indirectly associated with enhanced CCR2 levels and migratory activity in circulating monocytes. Additionally, CCR2 levels and chemotaxis activity had been reported to be significantly elevated in circulating monocytes in obese subjects (Krinninger P. et al., J Clin Endocrinol Metab 2014;99:2500). As BMI was higher in the enlarged LA group in both analyses of circulating and local monocytes/macrophages, higher BMI might be another possible cofounder in this study.  

Reviewer #2: Summary: Iowata et al examine the association of monocytes was structural remodeling of the left atrium, by examining blood samples from hundred 61 patients with atrial fibrillation. There a specific examination of chemo kind receptor protein levels, and chemoattractant related to monocytes. The authors conclude from the data shown that patients with enlarged left atria have higher level of monocytes/macrophages, and this presence may lead to higher rates of progression of atrial remodeling in this patient population. Overall, the rationale of the study is sound, the manuscript is well-written, and techniques and experimental approach are very thorough.

[Response]

We would like to thank the Reviewer for his/her careful review and the constructive comments on our manuscript. We have carefully revised the manuscript in response to the Reviewer’s comments and suggestions, which have greatly help to improve the contents of our study.

Questions and comments:

1. Regarding the type of specimen procured and examined: were there any potential confounding factors, such as any prior incidents of myocarditis or other forms of inflammation that would have altered monocyte/macrophage levels?

[Response]

We appreciate the Reviewer for raising an important point regarding any possible confounding factors to determine the phenotypes of monocytes/macrophages in AF patients. In the AF patients who underwent catheter ablation and whose circulating monocytes were examined, no participant had history of myocarditis; however, patients with enlarged LA had higher BMI, higher incidence of diabetes as well as history of heart failure, and lower ejection fraction compared to those with normal LA (Table1), all of which are associated with inflammation (Reilly SM. et al., Nat Rev Endocrinol 2017;13:633., Tsalamandris S. et al., Eur Cardiol. 2019;14:50., Adamo L. et al., Nat Rev Cardiol 2020;17:269). 

Additionally, in the population who underwent cardiac surgeries whose left atrial appendage were histologically examined, BMI was significantly higher in enlarged LA than normal group (Supplementary Table 1 in the original manuscript, now Table 2). 

A previous study enrolled non-obese and obese female showed that blood monocyte CCR2 expression and chemotaxis activity was significantly higher in obese subjects compared to those in lean subjects (Krinninger P. et al., J Clin Endocrinol Metab 2014;99:2500). Therefore, in AF patients, it is possible that high BMI might play a role for the higher migratory activity of circulating monocytes and higher number of monocytes/macrophages found in the LAA. Moreover, another study showed that, in patients with heart failure, abundance of CCR2-positive macrophages in the left ventricular (LV) tissue negatively correlated with LV systolic function, suggesting possible associations of cardiac function with the activity of CCR2-positive macrophages in the cardiac tissue (Bajpai G. et al., Nat Med 2018;24:1234). Accordingly, these factors, including higher BMI, history of heart failure and reduced systolic function, potentially affected CCR2 levels and migratory activity of monocytes in this study. We believe this is a very important discussion; therefore, we would like to add the following new discussion.

Revised Discussion (page 21, line 436-446)

Diabetes is associated with not only increased number of circulating monocytes (Min D. et al. Mediators Inflamm 2012;2012:649083), but also activation of inflammatory monocytes and macrophages (Flynn MC. et al., Front Pharmacol 2019;10:666). Moreover, a study enrolling patients with severe heart failure suggested a negative correlation between left ventricular ejection fraction (LVEF) and the abundance of CCR2-positive macrophage in the cardiac tissue (Bajpai G. et al., Nat Med 2018;24:1234). In our study, incidences of diabetes and heart failure were higher and LVEF was lower in patients with enlarged LA compared with normal LA group, indicating that these factors might be directly or indirectly associated with enhanced CCR2 levels and migratory activity in circulating monocytes. Additionally, CCR2 levels and chemotaxis activity had been reported to be significantly elevated in circulating monocytes in obese subjects (Krinninger P. et al., J Clin Endocrinol Metab 2014;99:2500). As BMI was higher in the enlarged LA group in both analyses of circulating and local monocytes/macrophages, higher BMI might be another possible cofounder in this study. 

2. Whether any sex-related differences in monocyte/chemo kind levels, considering the fact that there was a much higher proportion of atrial fibrillation patients were not only men, but also older?

[Response]

We thank the on-point question regarding whether there are any gender- and age-related differences in characteristics of monocytes and chemokine levels. Using the data from AF patients who underwent catheter ablation, we have compared monocyte characteristics and a chemokine level in men vs. women and >65 vs. ≤65 years old (Supporting information 1 and 2).

There was no significant difference between genders in respect of monocyte subsets (classical: p=0.88, intermediate: p=0.073, non-classical: p=0.18), CCR2 mRNA on monocytes (p=0.58), migratory activity of monocytes (p=0.69) and plasma MCP-1 levels (p=0.10) (Supporting information 1).

There are several reports regarding gender-related difference of monocyte subsets in healthy subjects, although results of these were inconsistent and still inconclusive (Hearps AC. et al., Aging Cell 2012;11:867, Jiang W. et al., PLoS ONE 2014;9:e114589). Moreover, in the diseased populations including AF, it remains largely unknown. A study examined CCR2 levels in monocytes showed that CCR2 expression in male-derived monocyte was higher compared to those derived from female (Sellau J. et al., Nat Commun 2020;11:3459). To our knowledge, no study so far examined gender-related differences of migratory activity of circulating monocyte.

Furthermore, we have divided AF patients who underwent catheter ablation into two groups by the median of their age (65 years old) to analyze age-related differences in monocyte characteristics and chemokine levels (Supporting information 2). No significant difference was observed between younger and older subjects in monocyte subsets (classical: p=0.62, intermediate: p=0.21, non-classical: p=0.71), CCR2 mRNA (p=0.67), migratory activity (p=0.56), and plasma MCP-1 levels (p=0.50). Similar to the gender-related differences discussed above, age-related difference in characteristics of circulating monocytes and chemokine levels have not been fully addressed, while several studies have evaluated subsets of monocytes (Hearps AC. et al., Aging Cell 2012;11:867) and MCP-1 level (Seidler S. et al., BMC Immunology 2010;11:30). In our study, it is less likely that gender and age clearly affected on the subsets and migratory activity of circulating monocyte, and MCP-1 level in AF patients; however, as the current exploratory study had small sample size, future study with larger sample size would address the gender- and age-related changes in monocyte biology. In light of this clinically important point raised by the Reviewer, we have inserted sentences in the Discussion section as follows.

Revised Discussion (page 20-21, line 429-435)

AF incidence is increased with age. Also male was reported to be more susceptible to AF, although the lifetime risk of AF was not different between male and female (Magnussen C. et al. Circulation 2017;136:1588). In our study, there was no gender- and age-related difference in characteristics of circulating monocytes in AF patients who underwent catheter ablation (data not shown). Studies in healthy subjects have suggested age- and gender-related changes of monocyte characteristics such as monocyte subsets and CCR2 levels in monocytes (Hearps AC. et al., Aging Cell 2012;11:867, Sellau J. et al., Nat Commun 2020;11:3459); however, more comprehensive analyses are necessary to fully understand the role of monocyte migration in AF pathogenesis. 

Supporting information 1 (review purpose only). Comparison between male and female with AF in monocyte characteristics and plasma MCP-1. A) Monocyte subsets, B) CCR2 mRNA levels in monocytes, C) migratory activity of monocytes, and D) plasma MCP-1 concentration were compared between male and female subjects who participated in the circulating monocyte analysis.

Supporting information 2 (review purpose only). Comparison between younger (<65 y.o.) and older (≥65 y.o.) subjects with AF in monocyte characteristics and plasma MCP-1 levels. A) monocyte subsets, B) CCR2 mRNA levels in monocytes, C) migratory activity of monocytes, and D) plasma MCP-1 concentration were compared between younger (<65 y.o.) and older (≥65 y.o.) subjects who participated in the circulating monocyte analysis.

3. Some of the potential confounding factors between comparing patients with atrial fibrillation versus non-AF could include diabetes mellitus (in and of itself and inflammatory state that could affect monocyte/macrophage levels independent of atrial fibrillation risk). How is this and other factors taken into account when interpreting the findings?

[Response]

We appreciate the valuable comments regarding the potential confounding factors. As the Reviewer pointed out, the enlarged LA group included more patients with diabetes, which could link with higher inflammatory status. The enlarged LA group indeed showed higher levels of hs-CRP compared to the normal LA group. A previous study found that monocyte number was increased in diabetic patients compared to non-diabetic subjects (Min D. et al., Mediators Inflamm 2012;2012:649083). Another study showed that chemotaxis towards vascular endothelial growth factor-A (VEGF-A) was attenuated in monocytes from patients with diabetes (Waltenberger J. et al., Circulation 2000;102:185), although few studies examined monocyte chemotaxis to MCP-1 in diabetes. These evidence suggested that diabetes might have affected migratory activity of monocytes in AF patients, therefore, we added the discussion regarding this point as stated in the response to the Reviewer’s Comment 1.

4. In Figure 5: please clarify how you elucidated that the infiltrating cells were monocytes, and subsequently differentiated into macrophages following infiltration. The sentence beginning on line 320 implies that is the case; if it is not the case, please reword the sentence to reflect the findings more accurately if needed.

[Response]

We appreciate the Reviewer for the opportunity to clarify this point and agree that it is not possible that CCR2 positive monocytes/macrophage cells were whether infiltrated or resident. A previous study examined macrophages in cardiac tissue suggested that CCR2-positive macrophages were maintained by both monocyte recruitment and proliferation, whereas CCR2-negative macrophages were replenished mainly by proliferation (Bajpai G. et al., Nat Med 2018;24:1234). It is possible that CCR2-positive macrophages found in the left atrial appendages in our study may also be infiltrated; however, we were unable to provide evidence to prove it. We have therefore replaced or weakened the expression of infiltration.

Revised Abstract Methods (page 2, line 39-41)

Second, to characterize the local monocytes in the atrial wall, the resected left atrial appendages (LAA) were histologically evaluated (n=20).

Revised section title in Results (page 15, line 318-319) 

Increased number of CCR2-positive monocytes/macrophages in the wall of left atrial appendages in the enlarged LA group

Revised Results (page 16, line 331-335)

Furthermore, the number of CCR2-positive monocytes/macrophages was also high in LAA resected from patients with enlarged LA, suggesting the possibility of enhanced infiltration of circulating monocytes into the left atrium, which subsequently differentiated into macrophages in AF patients with advanced atrial remodeling (Fig 5B, 5C mid panel)

Revised Figure 5 legend in Results (page 17, line 345-346)

Fig 5. Increased number of monocytes/macrophages in the wall of left atrial appendages in AF patients with enlarged LA.

Revised Discussion (page 18, line 365-367)

Finally, histological analysis in LAA surgically resected from AF patients confirmed that the number of CCR2-positive monocytes/macrophages was higher in AF patients with enlarged LA.

Revised Discussion (page19, line 394-397)

Our immunohistochemical analysis revealed that higher number of CCR2-positive and CCR2-negative monocytes/macrophages in LAA wall in AF patients with progressive atrial remodeling, which could possibly indicate enhanced infiltration of monocytes/macrophages. 

Revised Conclusion (page 22, line 462-474)

Enhanced migratory activity of circulating monocytes and increased number of monocytes/macrophages in the atrial wall may underlie LA remodeling and the pathophysiology of AF.

5. On line 331, please verify magnification – is it supposed to be 63x? It is listed as 630x.

[Response]

We apologize the Reviewer for causing the confusion. We used the confocal microscope equipped with 10x ocular lens and 63x objective lens, hence the total magnification was 630x. To describe the methods clearer, we have changed the sentence as follows.

Revised Figure 5 legend in Results (page 17, line 347-349)

A) Representative image of CD68 (green) and CCR2 (red) double-positive cells (monocytes/macrophages) at a high-power field magnification (63x objective) in a left atrial appendage. Scale bar: 5 μm.

Likewise, we have changed other sentences describing microscopy analysis to specify those values indicated objective magnification.

Revised Materials and Methods (page 8, line 163-165)

Fluorescence images were obtained from five random fields of view per well at 10x objective magnification (BZ-X700, KEYENCE, Osaka, Japan) and then the nuclei were counted using Fiji software[10], representing the cell number

Revised Materials and Methods (page 8-9, line 182-185)

The number of CD68-positive cells (monocytes/macrophages) and CCR2-positive monocytes/macrophages in 10 random field of view images (20x objective) was counted manually in a blinded fashion.

Revised Figure 4 legend in Results (page15, line 311-315)

A) Representative fluorescence images of migrated cells after incubation with or without 5 ng/ml of MCP-1 for 90 minutes (Scale bar: 100 μm, 20x objective). B) Number of nuclei was counted in fluorescence images obtained from five random fields of view per well at 10x objective magnification.

Revised Results (page 16, line 327-329)

We then quantified the average number of CD68+ and CCR2+ monocytes/macrophages in 10 random field of view images (20x objective) of LAA sections.

Revised Figure 5 legend in Results (page 17, line 349-354)

B) Representative images at low-power magnification (20x objective) in normal and enlarged LA patients. Scale bar: 10 μm. C) The number of CD68-positive cells (monocytes/macrophages) and CCR2-positive monocytes/macrophages, and percentage of CCR2-positive macrophages per randomly selected field of view at 20x objective magnification (normal LA: n=10, enlarged LA: n=10, mean ± S.D., unpaired t-test).

---

## [Editor Report · Decision Letter 1]

29 Sep 2020

Enhanced monocyte migratory activity in the pathogenesis of structural remodeling in atrial fibrillation

PONE-D-20-13849R1

Dear Dr. Iwata,

We’re pleased to inform you that your manuscript has been judged scientifically suitable for publication and will be formally accepted for publication once it meets all outstanding technical requirements.

Kind regards,

Eliseo A Eugenin, Ph.D.

Academic Editor

PLOS ONE

Additional Editor Comments (optional):

Dear Dr. Iwata

Thank you for submitting your paper to PLOSone. I have read the answers to the two reviewers and most of concerns are solve. The confusion of the patient groups analyzed is well explained and developed. A pleasure working with you and your group

Eliseo Eugenin

---

## [Editor Report · Acceptance letter]

2 Oct 2020

PONE-D-20-13849R1 

Enhanced monocyte migratory activity in the pathogenesis of structural remodeling in atrial fibrillation 

Dear Dr. Iwata:

I'm pleased to inform you that your manuscript has been deemed suitable for publication in PLOS ONE. Congratulations! Your manuscript is now with our production department. 

Kind regards, 

on behalf of

Dr. Eliseo A Eugenin 

Academic Editor

PLOS ONE